# Prepared for the expected but unready for the unexpected: Unmet distractor expectations slow braking responsiveness but improve lane-keeping precision in a virtual driving simulation

Andrea Massironi[1,2]*, Marco A. Petilli[1,2], Carlotta Lega[2,3,4], Simone Fontana[5,6], Emanuela Bricolo[1,2,4]

1 Department of Psychology, University of Milano-Bicocca, Milan, Italy, 2 Mind and Behavior Technological Center – MiBTec, University of Milano-Bicocca, Milan, Italy, 3 Department of Brain and Behavioral Sciences, University of Pavia, Pavia, Italy, 4 Milan Center for Neuroscience (NeuroMI), Milan, Italy, 5 School of Law, University of Milano-Bicocca, Milan, Italy, 6 Department of Informatics, Systems and Communication (DISCo), University of Milano-Bicocca, Milan, Italy

* a.massironi9@campus.unimib.it

## Abstract

Driving requires attentional control mechanisms to enhance the detection of driving-relevant objects and to mitigate interference from distractors. While distractor suppression can be proactively engaged when distracting events are expected, this mechanism also comes along with both costs and benefits, affecting responsiveness and accuracy even in their temporary absence. We explored the effect of distractor expectations on driving performance. Through an immersive driving simulator, participants (N = 24) performed an adapted version of the Distractor Context Manipulation (DCM) paradigm. They navigated a circuit and promptly pressed the brake pedal whenever a road sign-like target appeared, under three different conditions: a Pure Block without distractors, and two Mixed Blocks, featuring frequent irrelevant distractors (67% of trials) differing in perceptual complexity (Feature Search vs. Conjunction Search). We measured braking RTs and lane-keeping precision. Results showed that Mixed Blocks delayed braking RTs, even when distractors were temporarily absent, with the magnitude of this delay scaling with the perceptual similarity between target and distractors. Conversely, the same mechanism improved lane-keeping precision when neither targets nor distractors were present. Our findings suggest that distractor expectations modulate driving performance, entailing a trade-off between responsiveness and precision that depends on the characteristics of the distractor context.

## Introduction

Over the past few decades, advances in the automotive industry have significantly changed the way we navigate urban and suburban environments. The progressive

**Data availability statement:** "All relevant data are within the manuscript and its Supporting Information files.".

**Funding:** The current project has been conducted within the MUSA – Multilayered Urban Sustainability Action – project, funded by the European Union – NextGenerationEU, under the National Recovery and Resilience Plan (NRRP) Mission 4 Component 2 Investment Line 1.5: Strengthening of research structures and creation of R&D "innovation ecosystems", set up of "territorial leaders in R&D".

**Competing interests:** The authors have declared that no competing interests exist.

expansion of tightly interconnected road networks, together with cutting-edge driving technologies and modern road infrastructure, has made driving a ubiquitous human activity worldwide, while also posing significant challenges for public road safety policy. According to the *Global status report on road safety 2023*, published by the World Health Organization (WHO), road traffic accidents – primarily involving vulnerable road users – are recognized globally as the 12th leading cause of death across all age groups and, remarkably, the 1st among children and young adults [1].

Beyond speeding, **distraction** has been identified as a primary human factor contributing to road crashes in Western countries [2,3]. As outlined by the National Highway Traffic Safety Administration of the U.S. Department of Transportation (NHTSA), distracted driving occurs when the driver's attention is diverted from the primary driving task by other competing – even if potentially irrelevant – actions (e.g., responding to a phone call) or events (e.g., conspicuous billboards along the roadside) [3]. Importantly, distraction while driving may involve visual and/or auditory modalities [4], suggesting that specific perceptual and attentional processes are primarily engaged also at the onset of road accidents [4–9], thereby providing a foundation for cognitive and vision science to address this issue [10–13] (for all definitions, see S3 File. Glossary in the *Supporting* section).

Driving is a complex visuomotor activity that requires the integration of multiple sensory inputs, including visual, auditory, somatosensory, and vestibular stimuli. It also demands the coordination of sensory functions with cognitive operations, such as peripheral vision, gaze pooling, guidance, spatial memory, and navigation [14–17]. The primary role of cognition in driving has also been shown by consistent findings showing that the assessment of attentional functions (e.g., **selective attention**) predicts driving performance across multiple driving-related parameters [18].

Importantly, to effectively orchestrate these processes, the recruitment of cognitive control mechanisms is also necessary [14,19–22]. **Cognitive control** is defined as the ability to effectively integrate and coordinate one's thoughts and actions with the ultimate aim of efficiently achieving a behavioural goal [22]. Specifically, in the context of driving, managing the sensory complexity of the external environment is crucial for ensuring efficient navigation. Our cognitive system must optimize the processing of driving-relevant objects (i.e., other cars, pedestrians, traffic lights, etc.) while actively suppressing distracting information (e.g., billboards). Such control may be exerted either reactively or proactively [22,23].

**Reactive control** is a behavioural adjustment triggered by a distracting event and occurring immediately thereafter. For example, when driving in a metropolis such as New York, a driver's attention may be automatically captured by the highly conspicuous billboards on the skyscrapers of Times Square – an effect known as attentional capture [24,25]: in such a scenario, reactive control enables the driver to disengage from these distracting objects and redirect attention back to the ongoing driving task.

Conversely, **proactive control** anticipates interference from distracting objects or events, minimizing their impact before they occur. It is achieved through the sustained maintenance of behavioural goals as active representations that effectively

guide perception and action; therefore, it capitalizes on one's expectations and prior experience [22]. For example, prior experience of distraction from billboards may enhance the driver's ability to filter out such interfering information before their attention is captured, thereby allowing focus on other driving-relevant objects (e.g., other cars, pedestrians, traffic lights, etc.).

Reactive and proactive control processes have been extensively investigated using the **visual search** paradigm [23,24,26–32]. In visual search, and more specifically in detection search tasks, participants are asked to detect and respond to a relevant object – the target – among irrelevant objects – the distractors [33–35]. It has been shown that visual search tasks with high-frequency distractors trigger proactive distractor filtering processes [23,36]. When distractor features are known in advance, such processes may benefit from negative attentional templates – a representation of the distractor held in working memory – which improve search efficiency by guiding attention away from distractor features [37] (see also [38] for complementary evidence on how distractor expectations bias target templates). Neural signatures of such negative templates have revealed a robust engagement of proactive control processes prior to search onset [39].

Methodologically, the effects of control processes driven by distractor expectations have been investigated throughout the **Distractor Context Manipulation** (**DCM**) paradigm [40–43]. In this paradigm, participants detect and respond to a target object either in Pure Blocks, which are entirely free of distractors (TP-DA trials), or in Mixed Blocks, where TP-DA trials – physically identical to the former – occur alongside frequent trials involving irrelevant distractors. The unique distinction between the critical TP-DA trials of the two blocks lies in the heightened expectation of distractor occurrence, elicited by encountered distracting events in the Mixed Blocks. This expectation leads to the engagement of control mechanisms that proactively prepare for incoming distractors, unlike in Pure Blocks, where such expectations are absent. This design allows to detect and quantify the costs and benefits associated with distractor expectations by comparing performance on physically identical TP-DA trials across blocks with and without such expectations (i.e., Mixed vs. Pure Blocks) [40–43].

A typical finding from this comparison is the presence of a distractor expectation cost, manifested as slowed response times in TP-DA trials of the Mixed (vs. Pure) Blocks. This suggests that the proactive preparation of mechanisms to suppress distractors introduces a cost in responding to the target object when these mechanisms are primed but not actively engaged [40,43].

They also involve a speed-accuracy trade-off [44], where, despite slower response times, participants generally exhibit improved accuracy in their performance [43]. This trade-off typically arises in contexts where the negative consequences of imprecision outweigh those of slower responses, leading subjects to use more conservative strategies, like taking additional time for detailed information processing before producing a response [44,45].

Prior studies have shown that the cost-benefit ratio is modulated not only by the frequency of distractive events across blocks, which determines the incentive to suppress the interfering objects [40,46], but also by the specific type of distracting events, which imposes low vs. high cognitive demand [43]. Petilli et al. [43] – by integrating the DCM paradigm with a visual search task – explored the effect of top-down proactive control by systematically varying the type of distractors. Specifically, by varying the distractors' defining features, they designed two distinct Mixed Blocks, intended to elicit visual search strategies of different complexity, naming them **Feature Search** vs. **Conjunction Search**, respectively.

In Feature Search, the target object lacks shared basic features, such as color, with distractors. The distinctiveness of the target's color acts as a salient cue that can make targets "pop-out" from the visual scene, allowing for immediate detection without the need for detailed attentional processing. Therefore, it can be exploited to quickly "guide" attention towards the target, while minimizing interference from non-target objects. For example, a red object can be readily identified among an array of all blue ones.

Conversely, in Conjunction Search, targets and distractors share basic features, necessitating a cognitively demanding and serial examination of individual items [33,34,43] to determine whether the attended object matches the target. A typical example is searching for a red circle within an array of blue circles and red diamonds.

Interestingly, contrasting TP-DA trials' performance between Pure and Mixed Blocks, and distinguishing between Feature and Conjunction searches, revealed that the cost of distractor expectation depended on the distractor's perceptual features, with TP-DA trials in the Conjunction Mixed Block yielding slower RTs than those in the Feature Mixed Block [43].

In driving, the key role of expectations is well established, highlighting drivers' capacity to anticipate future paths and events through visual perceptual sampling, pre-established attentive sets, and predictive mechanisms [16,47–51]. To the best of our knowledge, only four studies have examined control processes recruitment in driving [6,52–54]. In these studies, participants drove in highly controlled simulators in which researchers systematically manipulated conditions – such as curving roads or simulated crosswinds – to probe reactive vs. proactive control. Behavioural measures were continuously recorded to assess driving performance, while oscillatory brain activity was concurrently monitored to link neural dynamics with anticipatory or reactive responses. Findings provided evidence for both reactive and proactive strategies at play while driving. Specifically, proactive driving has been shown to operate in advance of driving-relevant events, coordinating the driver's responses to different environmental demands and promoting a more strategic and adaptable allocation of attentional resources [53].

However, these studies emphasize only the motor and cognitive load aspects of the driving task. Specifically, they focus on vehicle control parameters – such as steering wheel error, vehicle heading error, driving lane variability and lane keeping – as influenced solely by non-visual environmental interference (e.g., curving roads or simulated crosswind). Therefore, they offer limited insight into how distractor expectations mechanisms affect performance during visuomotor tasks that integrate visual search alongside the primary motor driving task.

The main objective of the present study is to investigate distractor expectations mechanisms within a naturalistic-like, yet experimentally controlled task, that is, a virtual simulated driving task incorporating visual search through an adaptation of the DCM paradigm.

Participants were instructed to navigate a simulated driving environment while promptly responding by braking to a task-relevant object (the target) and concurrently ignoring task-irrelevant objects (the distractors) appearing within their driving field of view. Specifically, they completed three different blocks: a Pure Block, in which distractors were not expected, and two Mixed Blocks (Feature vs. Conjunction), in which distractors of varying perceptual similarity to the target were expected.

Based on previous findings, the working hypotheses are as follows: (i) braking response initiation would be delayed (i.e., slower RTs) in target-present/distractor-absent (TP-DA) trials of both Mixed-Feature and Conjunction Blocks as compared to the corresponding trials of the Pure Block; (ii) the magnitude of this effect would depend on the complexity of the visual search imposed by distractor features, with slower braking RTs expected in the Mixed-Conjunction Block relative to the Mixed-Feature Block. Additionally, for exploratory purposes, we measured lane-keeping, as an overall indicator of driving accuracy.

## Materials and methods

### Participants

Twenty-four neurologically healthy young adults – university students – (13 females and 11 males; mean Age: M = 26.37 years, standard deviation SD = 2.71 years) took part in the experiment voluntarily. They were recruited from April 21st, 2023, to July 14th, 2023, through the University Sona System – an online platform used for participant recruitment – in exchange for credits.

Most of the participants were right-handed (N = 19, i.e., 79.16% of the entire sample), and all reported normal or corrected-to-normal visual acuity and unimpaired color vision. Inclusion criteria comprised being aged between 21 and 35 years and holding a regular driving license for at least three years (M = 7.83 years, SD = 3.08 years).

We also administered a post-experiment survey in order to collect data on participants' driving habits. On average, participants reported driving 4.54 times per week (Min = 0, Max = 20; SD = 4.76) and covering an average distance of 45.50

km per week (Min = 0 km, Max = 140 km, SD = 44.20 km). Only four participants reported that they were not actively driving at the time of the experiment.

We estimated the required sample size based on the experimental work of Marini et al. [40], who employed the same paradigm as in the present study. Accordingly, based on a medium-to-large effect size of .30 [55], we performed a power analysis using the GPower software (GPower 3.1.7; [56]). The analysis indicated a minimum sample of 20 participants for F-tests (within-subjects ANOVA), with a conventional α = .05 and Power = .80. Nevertheless, we planned to test a total of 24 participants to ensure that each of the six possible orders of the blocks in our experimental design (see the *Stimuli and task design* section below) was assigned an equal number of times.

The protocol was reviewed and approved by Ethics Committee of the Department of Psychology at the University of Milano-Bicocca (Ethical Committee Prot. RM-2022–530). Written informed consent was obtained prior to participation, and participants were treated in accordance with the Declaration of Helsinki.

## Virtual driving simulation

The virtual driving simulator setup [9,57] was powered by a Nvidia Geforce RTX 3080 and consisted of three 24" FHD widescreen monitors (HP, 1920 x 1080) arranged in a semi-circular configuration, with the lateral monitors angled at 128° relative to the central one, to realistically reproduce both central and peripheral fields of view. Because the distance between participants and the simulator monitors was not fixed (see below), the subtended visual angle varied accordingly. On average, the setup simulated a horizontal field of view of 90.8° of visual angle.

To simulate the car's controls, we used standard racing game equipment, i.e., a steering wheel and a pedal set (Logitech G29 Driving Force). The steering wheel was placed at a distance of 53 cm from the central monitor. The driving seat was a standard office chair whose distance from the pedal set was independently adjusted – if needed – by the driver himself.

The simulation was programmed and run, and data were collected with the CarnetSoft driving simulation software (CarnetSoft®, version 7.1, 29th December 2020), which had already been successfully employed in the study of visual attention and distractor inhibitory mechanisms while driving [9].

Importantly, in line with previous studies using similar driving paradigms (see [6,9]), we set precise constraints on specific driving-related parameters to comply with the need for experimental control. First, the minimum and maximum vehicle velocity were fixed at 50 and 80 km/h, respectively, with an acceleration rate of 6 m/s$^2$. Furthermore, we set specific pressure thresholds to pedals for response recording in order to avoid possible confounders due to incidental, not goal-directed movements: specifically, the simulator returned – as output – that a braking response was initiated whenever the gas pedal was pressed less than 30% of its maximum range; similarly, an exerted pressure on the gas pedal greater than 70% of its maximum range was indicative of an actual acceleration response.

Data sampling frequency has been set at 100 Hz.

The virtual environment consisted of an extra-urban two-lane road within a daytime rural landscape, with a mostly sunny sky and a few scattered clouds. Drivers experienced a right-hand traffic (RHT) first-person racing perspective. The road environment included task-irrelevant lateral road signs and other vehicles travelling in the same direction. Audio-visual stimulation consistent with the environment was provided throughout the simulation to render the driving experience engaging and realistic. The road circuit was a one-way semi-circular loop, almost entirely rectilinear, with sharper bends at its extremities, allowing the drivers to re-enter the route (see Fig 1b).

Notably, although the described approach does not aim to reproduce all aspects of real-world driving and involves certain constraints typical of controlled simulated environments, it nevertheless allows for the investigation of cognitive processes relevant to driving under relatively ecologically valid, experimentally controlled and feasible conditions. This is an ideal compromise similarly adopted in numerous other studies investigating comparable cognitive processes [4,6,7,9,52–54].

**a.**

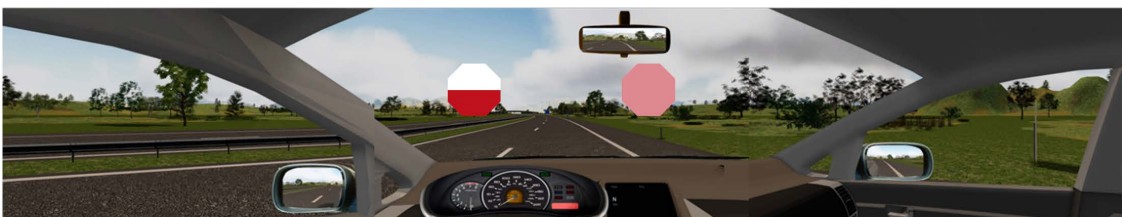

**b.**

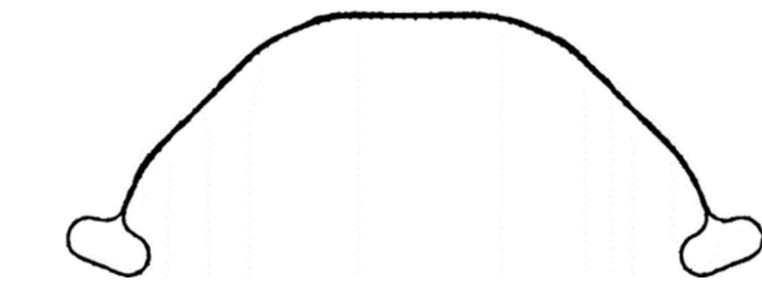

**Fig 1. (a)** Drivers' first-person racing perspective. In the image, an example of a Target stimulus (the white/red octagon) and a Distractor stimulus (the pink octagon) **(b)** Schematic representation of the road circuit.

The experiment was conducted within a darkened, controlled university laboratory.

### Stimuli and task design

Participants completed a visual search task, specifically a detection task [26], in which they were instructed to indicate whether a target object was present within their field of view while driving.

The experimental stimuli – adapted from VanRullen et al. [58] and used in Petilli et al. [43] – consisted of octagon figures rendered to resemble the disk part of typical vertical road signs. All stimuli had a vertical and horizontal dimension of 3.7 cm and 4 cm, respectively.

The target stimulus – for all blocks – (see Fig 1a) was a red-white octagon, with upper and lower colored halves defined by a bisection of its horizontal axis (RGB red: 192, 17, 30; RGB white: 255, 255, 255). To rule out possible facilitatory effects attributable to color position, the color orientation of the target was counterbalanced across subjects and remained the same throughout the experiment.

In the Pure Block, no distractors were ever present. In the Mixed-Feature Block, the distractor was a uniformly pink-colored octagon (RGB pink: 223, 136, 144, obtained by mixing the red and white RGB models of the target stimulus), while in the Mixed-Conjunction Block, the distractor was equivalent to the target, thus a horizontally bisected red-white octagon (RGB red: 192, 17, 30; RGB white: 255, 255, 255), but with reversed color orientation, as compared to the latter.

All stimuli were presented – for up to 2 sec – superimposed on the ongoing driving simulation, with a fixed eccentricity, notwithstanding the traveled distance and field of view depth. Stimuli were always displayed on the central monitor. More specifically, they were horizontally displayed leftward (4.3 cm from the left side of the screen) or rightward (4.3 cm from the right side of the screen) with respect to the midline of the central monitor and at a fixed vertical position of 6.9 cm from the upper side of the monitor, on a position where vertical road signs are typically displaced.

Critically, within the Pure Block in half of the trials (n = 30, i.e., 50%) only the target – and no distractor – was displayed (TP-DA trials), while in the other half of the trials (n = 30, i.e., 50%) neither the target nor the distractor was displayed (TA-DA trials).

Similarly, also the two Mixed-Blocks (i.e., Feature vs. Conjunction) comprised TP-DA trials (n = 30; ~16.67%) and TA-DA trials (n = 30; ~16.67%). However, Mixed-Blocks also included trials (n = 60; ~33.33%) where both the target and the distractor were displayed (TP-DP trials) and trials (n = 60; ~33.33%) where only two distractors were displayed, one for each visual hemifield (TA-DP trials).

A representation of the trial structure for each block is schematically depicted in Fig 2.

As can be noted, in the Pure Block there is a 0% probability that, should a stimulus appear, it would be a distractor (i.e., TP-DA trials), thus eliminating any kind of uncertainty about the expectation of what might happen in the next trials. On the contrary, in both Mixed Blocks, there is a ~66.67% (which is given by TP-DP trials + TA-DP trials) chance that a distractor will be displayed and just a ~16.67% that only the target will be on screen. As a result, in Mixed Blocks, a high expectation of distractor appearance in subsequent trials is generated.

Short breaks were given between each Block and within the Mixed Blocks every 60 trials. Accordingly, each experimental session consisted of 60 Pure, 180 Mixed-Feature, and 180 Mixed-Conjunction trials, for a total of 420 trials. Each trial lasted about 6/7 seconds; therefore, the pure Block had a duration of ~ 7 minutes, while Mixed Blocks of ~ 21 minutes

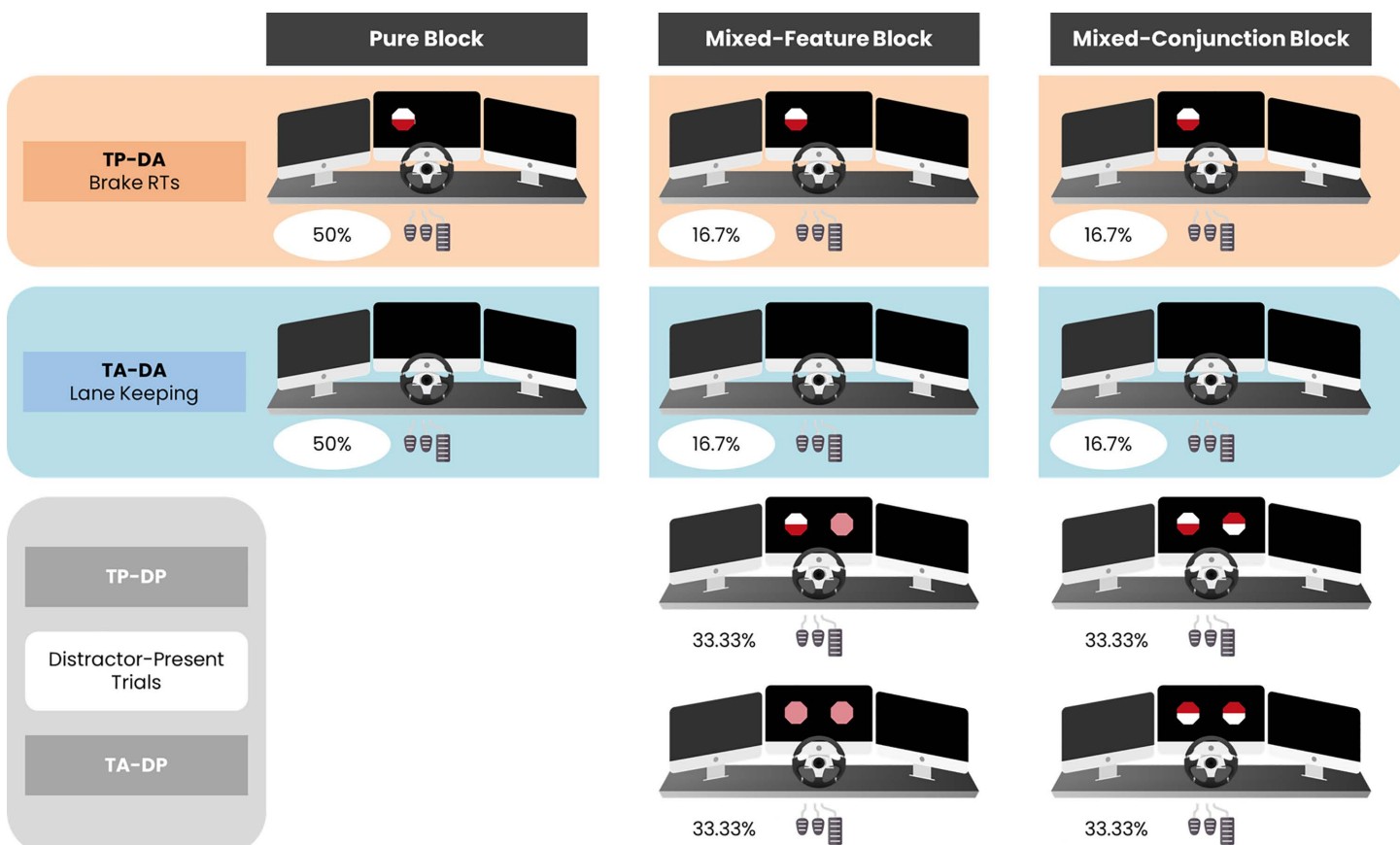

**Fig 2. Experimental Design.** The experimental task comprised three Blocks: The Pure Blocks consisting of 50% of Target Present-Distractor Absent trials (TP-DA) and 50% of Target Absent-Distractor Absent trials (TA-DA). Mixed-Feature (i.e., implying a feature visual search strategy) and Mixed-Conjunction (i.e., implying a conjunction visual search strategy) Blocks consisted, conversely, of a majority of Distractor Present trials (TP-DP and TA-DP), which outnumbered on a ratio of 4:1 (66.7%) the number of TP-DA (16.7%) and TA-DA (16.7%) trials. Critically, both TP-DA and TA-DA trials were structurally equivalent across Blocks and differed only in whether they belonged to a Block inducing distractor expectations or not (i.e., Mixed or Pure, respectively). Importantly, Braking RTs were measured in TP-DA trials, while Lane-Keeping was measured in TA-DA trials.

each, for a total driving time of ~ 49 minutes. Within-Block trial order was randomized but kept fixed across participants. Furthermore, the order of the Blocks was counterbalanced across participants following a complete counterbalancing procedure, which allowed us to assign each of the six possible orders an equal number of times (see also *Participants* section above).

## Procedure

Similar to previous studies (see [9]), participants completed a three-minute pre-experiment training session. The training session was aimed to familiarize participants with the driving simulator setup and, specifically, at allowing them to adapt their movements to the sensitivity of the steering wheel and the pedal set. No visual search task was included in the training. Moreover, the training could be repeated if additional familiarization was needed, but this was never required. During the experiment, participants were asked to "drive by remaining within the right lane, feeling free to follow the preferential trajectory within it". While driving, they were instructed to detect – whenever present – and respond – by quickly pressing the brake pedal – to the presence of the target stimulus, irrespective of any other distractor displayed. Both speed of response and accuracy were stressed. Throughout the entire experiment, participants were asked to always accelerate fully. Indeed, this ensured that all participants in all blocks consistently maintained the same speed (set at a maximum of 80 km/h) throughout the task (except after the braking response) and had the same range of movement to respond to the task-relevant target, from releasing the gas pedal to pressing the brake one.

Each experimental block was preceded by a one-minute training to let participants gain confidence with the current block, which allowed us to check for proper understanding of the instructions. At the end of the driving session, participants were ultimately debriefed about the study's aim.

## Statistical analyses

Braking reaction times (RTs) and Lane-Keeping were measured as our dependent variables of interest. Specifically, Braking RTs were operationalized as the time between stimulus onset and gas pedal release (pressure < 10%, see *Virtual driving simulation* section above). Lane-Keeping was operationalized as "Lane-Keeping Variability", defined as the standard deviation (SD) of the divergence of the front-wheel angles from the normative ones along the driver's ideal trajectory (i.e., the center of the right lane) within a time window of 6 seconds (see below).

For the "Lane-Keeping" variable, the use of the standard deviation (SD) is justified by the fact that it captures variability in maintaining the driver's preferred trajectory, irrespective of its position relative to the center of the right lane. In other words, as drivers were allowed to follow their own preferential trajectory within the right lane, lower variability – even despite significant differences in wheel divergence – would be an index of good lane-keeping performance.

Crucially, data on Braking RTs have been analyzed across all three blocks on physically identical TP-DA trials, while data on Lane-Keeping have been analyzed on TA-DA trials. More precisely, Lane-Keeping has been assessed within a time window of 6 seconds, comprising the 3 seconds immediately preceding the TA-DA trial onset and the 3 seconds immediately following it.

Importantly, data on Braking RTs were analyzed only for trials where participants responded accurately. This is a standard practice in overt visual search paradigms involving long exposure to stimuli, as these paradigms tend to capture variability in RTs more effectively than in accuracy. Indeed, for simple tasks, accuracy is usually less variable and typically reaches a ceiling level [59].

Statistical analyses on Braking RTs and Lane-Keeping have been conducted within the R environment (Version 4.5.1) [60], adopting Linear-Mixed Effect modeling [61] as the main statistical procedure through the "lmer" function of the "lme4" R package (Version 1.1.34) [62]. Block Type (three levels: Pure Block vs. Mixed-Feature Block vs. Mixed-Conjunction Block) and Road Curvature Variability – i.e., the SD of the variability in the road curvature (such as lane-keeping, assessed in a 6 seconds window spanning 3 seconds before and 3 seconds after TA-DA trial onset) – as well as the

interaction Block Type by Road Curvature Variability, have been entered in the model as fixed effects, while the by-subjects intercept was included as a random component. Statistical significance of main and interactive effects has been tested using the "anova" function of the "stats" package (Version 4.3.2) [60] through the F test and by adopting the Satterthwaite approximation equation for the computation of degrees of freedom. When needed, post-hoc tests were run using the chi-squared test in the R "phia" package (Version 0.2.1) [63], and the Holm method was applied to correct multiple comparisons. Furthermore, whenever the interaction Block Type by Road Curvature Variability turned out to be statistically significant, a trend interaction analysis was carried out using the "emtrends" function of the R "emmeans" package (Version 1.8.8) [64]. In the latter case, when needed, post-hoc tests were run by adopting the Kenward-Roger method for the computation of degrees of freedom and applying the Holm method for correction of multiple comparisons.

The complete dataset is available in the *Supporting Information* section (see S1 File).

## Results

### DCM accuracy

First, to assess correct task execution, we analyzed subjects' mean accuracy in braking responses to target presence across the three Block conditions (i.e., Pure Block vs. Mixed-Feature Block vs. Mixed-Conjunction Block). As expected, results indicate that in all three experimental Blocks, all subjects performed well above chance level, thus confirming an overall highly accurate task performance ($M_{Pure Block}$ = 0.99, Min = 0.93; $M_{Mixed-Feature Block}$ = 0.99, Min = 0.89; $M_{Mixed-Conjunction Block}$ = 0.98, Min = 0.87).

However, we still checked that there were no effects related to accuracy. We thus ran a generalized linear mixed model by entering accuracy as the dependent variable, Block Type and Road Curvature Variability as fixed effects, and the by-subject intercept as the random component. This analysis did not reveal any effect (all $p > .05$), likely due to the low variability of accuracy. Accordingly, in the subsequent analyses we focused only on RTs.

### Braking response

Braking RTs for correct responses were analyzed using raw data and, for consistency with prior studies [43], we report results based on these untransformed values. Notably, if log-transformed, the pattern of results did not change. The analyses on Braking RTs indicated a statistically significant main effect of "Block Type" [$F_{(2, 2079.1)}$ = 34.3, $p < .001$]. Indeed, post-hoc tests revealed that braking response times in the Pure Block were significantly faster ($EMM_{Pure Block}$ = 1.02 sec, $SE_{Pure Block}$ = 0.04 sec) compared to both the Mixed-Feature Block ($EMM_{Mixed-Feature Block}$ = 1.07 sec, $SE_{Mixed-Feature Block}$ = 0.04 sec) [$\chi^2_{(1)}$ = 27.4, $p < .001$] and the Mixed-Conjunction Block ($EMM_{Mixed-Conjunction Block}$ = 1.13 sec, $SE_{Mixed-Conjunction Block}$ = 0.04 sec) [$\chi^2_{(1)}$ = 152.9, $p < .001$] ones. Furthermore, braking responses in the Mixed-Feature Block were significantly faster than those in the Mixed-Conjunction Block [$\chi^2_{(1)}$ = 51.4, $p < .001$]. The latter result indicates that the magnitude of this effect is a function of the complexity of the specific type of distractor involved in the visual search. Finally, neither the main effect of Road Curvature Variability [$F_{(1, 2082.7)}$ = 0.3, $p = .568$] nor the two-way interaction Block Type*Road Curvature Variability [$F_{(2, 1987.2)}$ = 0.1, $p = .906$] turned out to be statistically significant.

To rule out the possibility that outliers might have biased the results, we subset our data to remove all observations with absolute standardized residuals > 2.5 SD relative to the data predicted by the model and re-run the same analyses as before. Accordingly, the output did not change after removing outliers (2.1% of the data), as we replicated the same pattern of results (i.e., main effect of Block Type $p < .001$, main effect of Road Curvature Variability and Block Type*Road Curvature Variability interaction $p > .05$) (Fig 3). Finally, to rule out the possibility that the four participants who reported not actively driving at the time of the experiment biased the overall results due to their lack of recent driving experience, we re-run all the analyses after excluding them from the sample. The pattern of results remained unchanged, with all main significant effects being replicated. This allowed us to confidently conclude that our sample was not biased by participants' current driving experience.

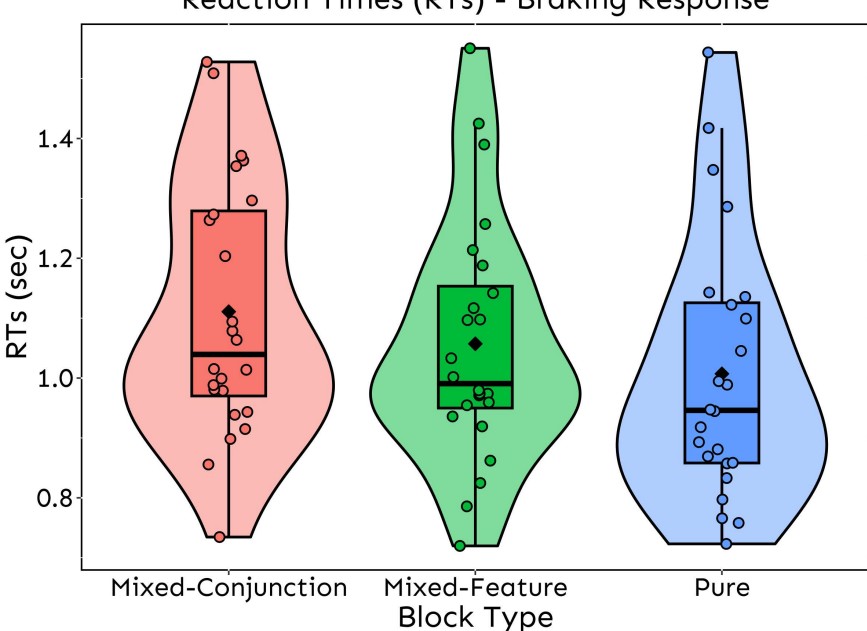

**Fig 3. Braking RTs.** Violin Box-Plot of braking response RTs as a function of Block Type after removing outlier observations. The graph shows the main effect of Block Type on RTs: distractor expectation entails a slowdown in braking response (i.e., Mixed Blocks RTs > Pure Block RTs); furthermore, the magnitude of this effect seems to be a function of the visual search complexity (i.e., Mixed-Conjunction Block RTs > Mixed-Feature Block RTs). Diamonds represent average RTs for each Block Type, while black lines represent the median; dots represent the individual average RTs.

## Lane-keeping

The analysis on Lane-Keeping did not reveal any effect of Block Type [$F_{(2, 2132.1)}$ =2.8, $p$ = .06], while it revealed a statistically significant main effect of Road Curvature Variability [$F_{(2, 2152.1)}$ = 28289.2, $p < .001$]. Furthermore – and more interestingly – the analyses revealed a statistically significant interaction between Block Type and Road Curvature Variability [$F_{(2, 2133.8)}$ =10.27, $p < .001$]. In this analysis, Road Curvature Variability had a negative impact on Lane Keeping across all blocks (i.e., the higher the Road Curvature Variability, the lower the accuracy in Lane-Keeping).

Notably, this effect was mitigated in both the Mixed-Feature ($b_{Mixed-Feature Block}$=0.688, $SE_{Mixed-Feature Block}$=0.006) ($t_{(2135)}$ = −4.103, $p < .001$) and the Mixed-Conjunction ($b_{Mixed-Conjunction Block}$=0.688, $SE_{Mixed-Conjunction Block}$=0.007) ($t_{(2134)}$ = −3.767, $p < .001$) Blocks as compared to the Pure Block ($b_{Pure Block}$=0.726, $SE_{Pure Block}$=0.007). Crucially, the Mixed-Feature vs. Mixed-Conjunction Blocks contrast turned out not to be statistically significant ($t_{(2133)}$ = −0.004, $p = 0.997$). Overall, these results indicate that distractor expectation mitigates the negative impact of driving demands on the trajectory accuracy, independently of the type of distractor expected. As for braking response analyses, to exclude the possibility that the presence of outliers biased the results, data have been sub-set to remove all observations with absolute standardized residuals > 2.5 SD relative to the data predicted by the model, and the analyses were then re-run on the new dataset. Accordingly, the output did not change after removing outliers (3.8% of the data), as we replicated the same significant interaction ($p < .001$); additionally, the main effect of Block Type became significant ($p < .001$) (Fig 4). Finally, to rule out any potential bias from participants lacking recent driving experience, we re-run all the analyses after excluding those who reported not actively driving at the time of the experiment. The pattern of results remained unchanged, allowing us to conclude that our sample was not biased by recent driving experience.

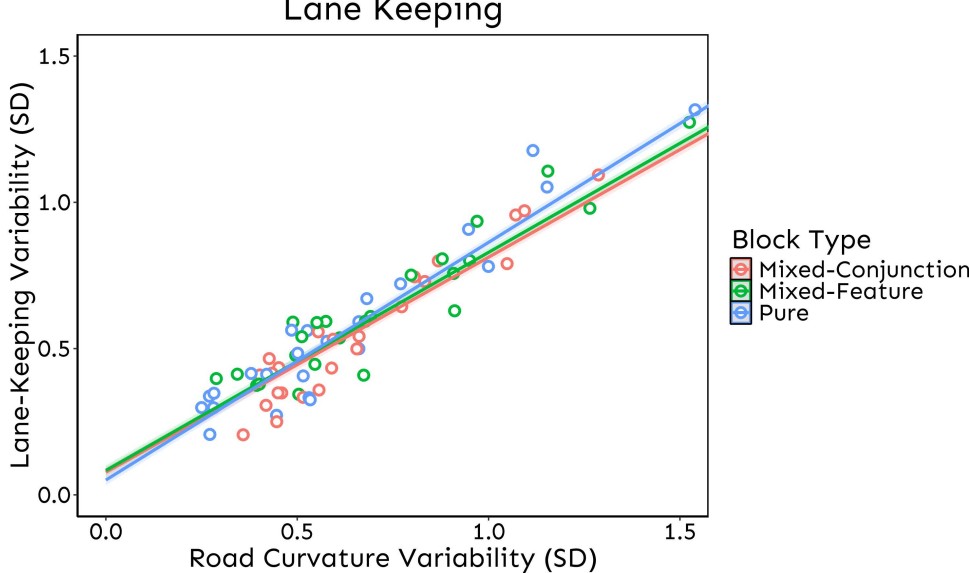

**Fig 4. Lane-Keeping.** Trendlines of Wheel Error as a function of Block Type and Road Curvature Variability after removing outlier observations. The graph shows the interactive effect between Block Type and Road Curvature Variability on Lane-Keeping Variability: distractor expectation entails a more accurate performance (i.e., Mixed Blocks Lane-Keeping Variability < Pure Block Lane-Keeping Variability) in maintaining the vehicle on path when the environmental demand is higher, i.e., at greater levels of Road Curvature Variability; notably, this effect appears to be independent of the visual search complexity (i.e., Mixed-Conjunction Block Lane-Keeping Variability = Mixed-Feature Block Lane-Keeping Variability). Each dot represents, for each subject and each condition, the mean of Lane-Keeping Variability as a function of the mean Road Curvature Variability.

## Discussion

The primary objective of the present study was to explore the effects of distractor expectation processes on driving performance using a naturalistic-like yet highly controlled driving scenario.

Notably, and in line with our hypotheses, we replicated the behavioural effects of distractor expectations found by Petilli et al. [43], this time in responses relevant to driving performance. Specifically, our findings indicate that the recruitment of distractor expectation processes, triggered by an uneven distribution of distractor occurrences, slowed drivers' braking responses to task-relevant targets. Importantly, this effect was observed even in the absence of potentially disruptive distractors, as evidenced by faster braking response times in the Pure Block compared to both Mixed-Feature and Mixed-Conjunction Blocks. Moreover, we showed that the magnitude of this effect depended on expected visual search complexity, with RTs in the Mixed-Feature Block being significantly faster than those in the Mixed-Conjunction Block. Ultimately, variability in road curvature was associated with better driving performance in maintaining optimal vehicle positioning during Mixed Blocks, where anticipatory control mechanisms for distractor expectation were engaged, relative to Pure Blocks, in which no distractors were expected. Our findings suggest that, irrespective of distractor type, the mere anticipation of distractors mitigates the disrupting effect of driving demands, thereby enhancing drivers' ability to maintain the optimal trajectory in Mixed Blocks as compared to the Pure Block.

Overall, our findings highlight the key contribution of distractor expectations processes – with both costs and benefits – in modulating driving performance, an everyday task in which visual search is an integral component.

Driving is a complex cognitive activity that relies on the integration of multiple visuomotor skills to achieve optimal navigational outcomes [14,15,17]. Specifically, to efficiently manage the perceptual complexity of crowded and dynamic environments, it is crucial for drivers to engage cognitive control processes, which enable the fine-tuned orchestration of these cognitive skills.

Cognitive control enables effective coordination of thoughts and actions and relies on the integrity and resilience of working memory processes against distractors, supporting the retention and maintenance of goal-relevant information as active internal representations that guide action during ongoing task performance [19,22,65].

In visual search, attentional control can be facilitated by negative attentional templates, which rely on memory representation of distractors to enhance search efficiency by guiding attention away from objects with distractor-matching features [37,39]. Moreover, search efficiency has been linked to evidence accumulation (EA) processes, in which perceptual information is progressively gathered until one of two mutually exclusive decision thresholds is reached, thereby supporting decision-making and enabling the execution of context-appropriate actions (e.g., Go/Brake – vs. No Go/No Brake) [66–68]. We hypothesized that the frequent occurrence of distractors in Mixed Blocks may have triggered the activation of task-sets related to distractor expectations – possibly as negative attentional templates – manifesting as a general enhancement in attentional processes sustained during driving. This mechanism may have influenced evidence accumulation by modulating drift rates, resulting in more cautious (slower) braking responses. This reflects the cognitive system's tendency to prioritize accuracy over speed in uncertain contexts where error costs are higher [44], impacting even distractor-free trials. However, such a more cautious task set in anticipation of increased cognitive demands may also underlie the observed improvement in lane-keeping performance. This hypothesis is also consistent with data from Exp. 4 of Petilli et al. [43], which demonstrated that the cost associated with distractor expectation is more likely attributable to a proactive shift toward a more conservative (i.e., cautious) response criterion rather than to distractor-filtering processes or inter-trial variability [43].

While firm conclusions about the precise mechanisms underlying our findings (e.g., the involvement of negative attentional templates) cannot be drawn from this speculation, we consider it is more likely that a proactive mechanism, rather than a reactive one, accounts for our data. This interpretation is supported by the analysis on braking RTs in TP-DA trials, where the absence of distractors rules out the involvement of reactive distractor suppression mechanisms. Furthermore, given that TP-DA trials were physically identical across blocks, any difference in RTs can be attributed to varying levels of distractor features and associated expectations. To address this open question, neural activity measures (e.g., EEG) would be beneficial, particularly through time-frequency analysis focusing on theta oscillations over frontoparietal regions, theta phase synchronization in prefrontal regions and posterior alpha power [39].

Notably, the slowing observed in the Mixed Blocks could alternatively reflect the higher frequency of prospective-event signals requiring stimulus evaluation, which may help maintain a sustained state of braking readiness throughout the block. In principle, such an increased rate could prolong the activation of this readiness state and, potentially, delay responses to actual braking signals. However, this account does not fully capture the pattern of findings reported across studies employing the same paradigm used here. Indeed, proactive slowing has been consistently observed across a wide range of laboratory tasks, even when the frequency of events requiring stimulus evaluation is matched across blocks (e.g., crossmodal visuo-tactile designs, [40]; flanker tasks, Marini et al. [40,41]; visual search, Experiment 4 in Petilli et al. [43]). These converging findings demonstrate that the observed cost cannot be attributed to a sustained braking schema driven by higher event pacing. Instead, it more likely reflects proactive control processes triggered by an increased expectation of distractors. In line with this interpretation, the present study shows that proactive-control costs associated to distractor expectations persist and translate into a characteristic pattern of driving behaviour when embedded within realistic driving simulations.

It could also be argued that our findings might alternatively be explained by differences in task demand and cognitive load, stemming from the adoption of different strategies to exploit the search task. Specifically, whereas Feature Search primarily relies on bottom-up perceptual distinctiveness between target and distractors [27,69], Conjunction Search – where target selection and/or distractor suppression are largely, though not exclusively, driven by top-down processing [70,71], engages more controlled mechanisms, thus making this type of search more cognitively demanding.

While this represents a plausible alternative framework, we believe it can be dismissed for three reasons: (i) if the modulation of our observed RTs were fully attributable to fluctuations in vigilance, we would expect RTs to progressively slow down over the course of a block, an effect referred to as vigilance decrement [72]. Furthermore, if vigilance levels are scaled as a function of cognitive demand, the rate of vigilance decrement should vary across Pure, Feature, and Conjunction Search blocks, as well as between TP-DA and TP-DP trials, because being physically different, they would likely elicit distinct responses. However, our data do not support this prediction (see S2 File. *SupplementaryMaterials* in the *Supporting Information* section); (ii) although our Braking RTs results suggest that the observed effect is modulated by the visual search complexity imposed by distractors features, it should also be noted that this analysis was conducted on TP-DA trials, where no distractors were present on-screen. Therefore, since the comparison across blocks was based on physically identical trials, the observed differences are unlikely to reflect varying task demands or reactive distractor suppression mechanisms, but rather differences in attentional sets linked to distractor expectations; accordingly, RTs differences might stem from varying levels of expectation about the trial *n* being more or less demanding, rather than from a pure effect of task demand; accordingly, if anything, it is the "expectations" of higher cognitive demand to impair performance when those are not met, an effect we hypothesize to be proactive, although additional methods (e.g., EEG) would be required to directly address this question; (iii) if task demand had truly affected driving performance, a negative impact should have been observed in Lane-Keeping for Mixed (vs. Pure) Blocks. Instead, we found that merely expecting distractors improved drivers' ability to maintain optimal vehicle positioning along the route. Significantly, our findings are consistent with key results from classical lab-based attention research [43], despite being obtained in a markedly different setting – namely, a driving simulation involving dynamic, context-rich visual environment and naturalistic driving responses, as opposed to standard laboratory tasks relying on basic visual displays and button-press responses.

In this regard, the lane-keeping findings represent a good example. Our data suggest that mechanisms activated by expectation of distractors mitigates lane-keeping deterioration under more demanding driving conditions, an effect clearly observable even in the absence of stimuli to detect or filter out (i.e., TA-DA trials). This indicates that the speed-accuracy trade-off extends to the overall driving task. In other words, the involvement of distractor expectation processes, while leading to slower responses – consistent to prior works [40,43] – appears to enhance driver's accuracy to maintain vehicle's control, even when no visual search demand is present.

Despite its contributions, our study still presents a relevant limitation that must be considered. In particular, while this study represents an important step toward scenarios that more closely approximate the perceptual and motor demands of real-world driving, it still involves a relatively high level of experimental control and artificial driving demands, which constrain the direct generalizability of our findings to real-world driving contexts. Vision science has long recognized the methodological challenge of modelling real-world driving conditions (see, for example, [16], for a review). Capturing its fundamental aspects can indeed be achieved through a continuum of approaches, from highly controlled laboratory experiments to naturalistic field studies, each reflecting a different balance between external validity and experimental control. It may thus be argued that our task does not reflect genuine driving performance, and that the driving component merely serves as a context for a standard lab-based experiment on the cognitive processes underlying distractor's expectations. However, our setup utilizes a markedly different approach from standard laboratory tasks [43], that is a driving simulation featuring a dynamic and context-rich visual environment. Additionally, the response measures are inherently naturalistic driving actions, whose motor demand differs significantly from button-press responses common in laboratory settings. Specifically, the first critical measure, timely braking in response to abrupt, context-dependent events, reflects a natural, safety-relevant action which typically occurs only in specific scenarios (e.g., a child crossing the street, a cyclist running a red light, or a tree branch falling onto the road during heavy rain and strong wind), whereas the second, lane-keeping, is typically absent in standard experimental design. Even more importantly, lane-keeping was recorded and analyzed in the absence of any attentional and visual demand unrelated to driving. Ultimately, the use of driving simulations and motor responses that reflect real-world driving tasks has been recognized as a scientific standard, as evidenced by numerous

studies investigating attentional processes at the wheel [4,6,9,52–54,73]. While lane-keeping can be reliably assessed even in highly controlled settings, context-specific events vary in likelihood, and the way drivers act upon them is shaped by environmental cues and prior experience. Given this variability, and to ensure methodological rigor, we employed a generic driving environment with neutral stimuli that drivers had to learn how to respond. This approach allowed us to map onto general and fundamental aspects of driving performance that are likely generalizable across a wide range of real-world scenarios.

Collectively, these methodological aspects demonstrate that participants were fully engaged in a genuine driving-related task, with behavioral responses directly reflecting cognitive and motor demands inherent to real world-driving.

While our study prioritizes internal validity over external validity, thereby limiting direct and meaningful generalizability, we nonetheless believe that our data offer relevant insights when considered within the context of current driving and spatial navigation literature.

Predictability, uncertainty and expectation processes play a key role in road safety [16,48,50]. Moreover, given humans' remarkable ability to anticipate waypoints and unexpected events [49,74–81], future studies should systematically investigate how these processes affect driving performance and spatial navigation.

For example, an intriguing hypothesis to investigate more rigorously is that maximizing predictability may not be advantageous as excessive reliance on anticipatory mechanisms could impair driving performance when unexpected, yet task-relevant, events occur. Our findings suggest that the cost-benefit ratio between braking slowdown and lane-keeping improvement is not proportional: while costs progressively increase (reflecting the interfering effect of distractors), benefits remain constant. This raises the hypothesis that there may be a threshold where, if costs are minimized (i.e., by using distractors with an extremely low interfering impact), their weight becomes marginal enough for benefits to outweigh costs.

The role of predictions and expectations in driving has also been considered in urban design, as exemplified by the "self-explaining roads" (SER) framework [11,82,83]. The SER framework suggests that anticipatory mechanisms in visual cognition should be considered in the design of roads and road signs, thereby leveraging top-down expectations to support optimal driving performance [11,82,83].

Although a detailed discussion of this issue is beyond the scope of the present paper, expanding this line of research could help clarifying challenging driving phenomena – for example, monotonous driving [73] and the "close-to-home" effect [84,85] – where reduced uncertainty may trigger adverse consequences through anticipatory mechanisms. Moreover, it could inform the development of road safety measures aimed at supporting these mechanisms within optimal limits, so as to balance their potential benefits and costs for driving performance.

## Conclusions

In conclusion, our findings show that distractor expectations negatively impact braking response times to task-relevant objects, even when distractors are absent. At the same time, merely expecting distractors enhance drivers' ability to maintain optimal vehicle positioning, even when no visual search demand is present and driving demands are high.

Overall, our findings suggest that distractor expectation processes have a sustained impact on driving performance, with specific costs and benefits depending on the characteristics of the distractor context.

Future studies should more rigorously investigate the specific cognitive mechanisms underlying distractor expectations' impact on driving performance, as this line of research holds potential value for policymakers and practitioners focused on improving road safety, ranging from urban and road design and legislative initiatives to the development of advanced in-car driver assistance technologies.

## Supporting information

**S1 File. Data.**
(CSV)

**S2 File. SupplementaryMaterials.**
(PDF)

**S3 File. Glossary.**
(PDF)

## Author contributions

**Conceptualization:** Andrea Massironi, Marco A. Petilli, Carlotta Lega, Simone Fontana, Emanuela Bricolo.

**Data curation:** Andrea Massironi, Marco A. Petilli.

**Formal analysis:** Andrea Massironi, Marco A. Petilli, Carlotta Lega.

**Investigation:** Andrea Massironi.

**Methodology:** Andrea Massironi, Marco A. Petilli, Carlotta Lega, Simone Fontana, Emanuela Bricolo.

**Project administration:** Marco A. Petilli, Emanuela Bricolo.

**Software:** Simone Fontana.

**Supervision:** Emanuela Bricolo.

**Visualization:** Andrea Massironi, Marco A. Petilli.

**Writing – original draft:** Andrea Massironi.

**Writing – review & editing:** Andrea Massironi, Marco A. Petilli, Carlotta Lega, Simone Fontana, Emanuela Bricolo.

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
