## [Decision Letter · Decision Letter 0]

30 Apr 2025

Dear Dr. Massironi,

Editor comment. I was able to secure two reviewers, both of whom have now provided their evaluations and comments. One conceptual issue that particularly strikes me is the following: while the manuscript frames the study as examining driving performance or aspects of driving, it appears to me that the present task does not genuinely involve driving performance in any substantive sense. Rather, it seems to be a standard cognitive experiment where driving serves merely as a superficial cover story without any genuine relation to the behavioural and cognitive demands of actual driving. I do not refer here to issues of ecological validity or to the need for more realistic settings; rather, I question whether the task meaningfully engages with the construct of driving at all. I would encourage you to reflect on this issue carefully in your revision, particularly with regard to how you frame the purpose and implications of your study. I invite you to prepare a revision of your manuscript, accompanied by a response letter in which each comment is addressed individually in a point-by-point manner.

I look forward to receiving your revised submission.

Yours sincerely,

Michael B. Steinborn

Section Editor, Plos ONE

We look forward to receiving your revised manuscript.

Kind regards,

Michael B. Steinborn, PhD

Section Editor

PLOS ONE

Journal Requirements:

2. Please include captions for your Supporting Information files at the end of your manuscript, and update any in-text citations to match accordingly. Please see our Supporting Information guidelines for more information: http://journals.plos.org/plosone/s/supporting-information .

Reviewers' comments:

Reviewer's Responses to Questions

**Comments to the Author**

1. Is the manuscript technically sound, and do the data support the conclusions?

Reviewer #1: No

Reviewer #2: Yes

2. Has the statistical analysis been performed appropriately and rigorously?

Reviewer #1: Yes

Reviewer #2: Yes

3. Have the authors made all data underlying the findings in their manuscript fully available?

Reviewer #1: Yes

Reviewer #2: Yes

4. Is the manuscript presented in an intelligible fashion and written in standard English?

Reviewer #1: Yes

Reviewer #2: Yes

Reviewer #1: This manuscript explores the effects of proactive attentional control on two aspects of driving behavior—braking and lane-keeping—within a simulated environment. By adapting the Distractor Context Manipulation (DCM) paradigm to a driving context, the authors show that distractor expectations can slow down braking responses while improving trajectory stability. While this study is innovative and features an original design, the interpretation regarding the role of proactive control remains limited, as this mechanism is not directly manipulated by the authors and thus remains speculative. A substantial revision of the introduction is recommended prior to acceptance.

- The introduction is generally well written and well referenced. The authors appropriately cite the work of Nick Gaspelin and J Geng. However, the introduction would benefit from citing recent work on negative templates in visual search (i.e., representations of distractors in working memory), particularly the contributions of Nancy Carlisle (Negative and Positive Templates: Two Forms of Cued Attentional Control, 2023) and its relevance to the recruitment of proactive control for distractor suppression (Chidharom & Carlisle, 2023, Journal of Cognitive Neuroscience).

- Do vigilance levels differ across blocks in terms of accuracy and response speed? One might imagine that the Mixed-Conjunction block, being more cognitively demanding, taxes sustained attention more heavily, potentially explaining slower RTs.

- The cognitive demand—and thus mental effort—is higher in the Mixed-Conjunction block. Therefore, it is plausible that the slowing of RTs is not due to modulation of proactive control per se, but rather reflects increased general attentional engagement, which could also account for the improved lane-keeping performance.

- In the introduction, the authors cite literature on both proactive and reactive suppression of distractors. However, their paradigm does not allow for a direct manipulation of these two mechanisms. It is entirely possible that, due to its similarity with the target, the distractor in the Mixed-Conjunction block captures attention, and that the observed cost reflects a reactive suppression process that reallocates attention to the target.

- To support a claim of proactive control being engaged to suppress distractors, physiological evidence (e.g., EEG or eye-tracking) would be necessary—see, for example, the work by Nick Gaspelin.

- Behaviorally, could the authors show that, as distractors repeat, reaction times to detect the target become progressively faster? This would support the idea of learned suppression (Gaspelin & Luck), which is believed to rely on proactive control mechanisms.

- Overall, I would recommend that the authors revise their manuscript not around the concept of proactive control per se, but rather more broadly around distractor suppression and its interference with driving performance. As currently presented, the evidence for proactive control is not sufficiently convincing to warrant such strong claims.

Reviewer #2: The subject of this study is quite engaging, and I have several recommendations to enhance its quality:

1. The simulation employs Carnet software. It would be advantageous for the authors to incorporate more details regarding the road environment, weather conditions, and traffic scenarios.

2. I recommend verifying the standards used for the testing process. For example, the simulation speeds are set at 50 and 80 km/h. What criteria were considered in selecting these specific speeds?

3. In my view, a three-minute training session for new users is inadequate, as the simulator's operation significantly differs from that of a real vehicle. Please offer your justifications for this training duration.

**Do you want your identity to be public for this peer review?** For information about this choice, including consent withdrawal, please see our Privacy Policy

Reviewer #1: No

Reviewer #2: No

---

## [Author Response · Author response to Decision Letter 1]

24 Sep 2025

Please see the attached document 'Response to Reviewers', where we provide detailed, point-by-point replies to all comments from the Editor and Reviewers.

---

## [Decision Letter · Decision Letter 1]

13 Oct 2025

Dear Dr. Massironi,

We look forward to receiving your revised manuscript.

Kind regards,

Michael B. Steinborn, PhD

Section Editor

PLOS ONE

Journal Requirements:

Additional Editor Comments:

Background

This is a kind of dual-task study conducted in the context of simulated driving, where a primary task required participants to steer a car with a Logitech wheel and pedal simulator on a curved two-lane road, and a secondary task required them to watch for a target (a standard red stop road sign), and to press the brake pedal as soon as it appeared. Three conditions were tested, in the first, only the normal target (stop road sign) appeared, and participants where to brake whenever it was shown, in the second, the real stop sign was mixed with colour-altered signs that looked similar but were not true stop signs, and in a third condition, the real stop sign was mixed with more complex variations, where the colours and layout were changed, making them visually confusing but still non-targets. In every case, participants were instructed to brake only for the standard red stop sign and to ignore all others. The results showed that when these altered signs appeared frequently in the inter-mixed conditions, participants became slower to brake even when the true stop sign appeared, but at the same time they kept the car more precisely within the lane. This indicates that expecting possible distraction delays the braking response yet improves the readiness for steering control.

Evaluation. 

This study is clearly a laboratory-based experiment, and as such it represents a controlled but necessarily simplified model of real driving. The simulated setting does not reproduce all aspects of driving a real vehicle, which limits the degree of ecological realism. At the same time, it is evident that modelling genuine driving is inherently difficult, as one must decide which features of the task are critical to retain while keeping the situation experimentally manageable. In this respect, the present study achieves a reasonable balance. The simulator provides a physically embodied steering and braking task, which captures the continuous control demands of real driving while still allowing for precise experimental manipulation. The sample size (N = 24) is modest but acceptable for this kind of within-subject design, where each participant contributes a large number of trials. The manipulation is well structured and internally consistent: one condition involves responding only to the true stop sign, while the mixed conditions introduce non-target events that vary in visual similarity to the target. The essential logic of the design is sound, but see my detailed comments.

(1) some basics: abstract and title

Abstract: The abstract is nearly 300 words long, too wordy, and overly narrative. It contains several passages that are not typical for a scientific abstract and that obscure rather than clarify the main message. I recommend shortening it substantially, and restructuring it around four clear elements: what is aimed, what is done (rationale, design, factors, and measures), what is found, and what this means conceptually (the take-home message).

Title: The current title (Optimizing driving performance: the impact of distractor expectations on braking and lane-keeping) is somewhat misleading. The study does not involve optimisation of real driving, nor does it test distractor expectations in the applied sense implied by the title. Instead, it examines a dual-task situation combining a continuous primary task (background task: lane keeping) with a secondary prospective-event task (braking to stop road signals). I therefore recommend rewording the title to reflect this conceptual focus. Something short and memorable that conveys the “just do it when you get a chance” logic of the paradigm would better capture the actual content and originality of the study.

(2) Ecological validity

This is clearly a laboratory experiment, and it does not represent real driving in any strict sense. It uses artificial visual events that would never occur in real traffic, and it isolates them from the contextual cues that normally accompany such events. In reality, a stop sign appears at an intersection, surrounded by situational markers that jointly define its meaning. In the present simulation, by contrast, stop-sign stimuli appear at arbitrary positions along a straight segment, outside any traffic context. The manipulation therefore represents a situation that is not merely decontextualised but almost contra-situated relative to genuine driving. From the standpoint of experimental logic, this is legitimate, yet it shows that the factorial structure of the study is conceptually removed from the ecological conditions it nominally models. On the other hand, within the limits of a laboratory, the study is remarkably well implemented and technically accomplished. The use of a Logitech G29 wheel and pedal system provides continuous sensorimotor feedback, and the visual immersion is sufficiently rich to induce a sense of genuine control. In this respect, the study reaches the upper bound of what can be achieved within a behavioural-laboratory setting. It captures the feeling of driving through steering, braking, and lane keeping, even if the external events remain abstract.

Conceptually, the design follows the logic of a continuous dual-task paradigm, where a primary task (lane maintenance) is performed as background while a secondary task (braking to infrequent events) occurs intermittently. In this way, the structure parallels prospective-memory designs in which participants respond to an occasional cue “when they get a chance” while engaged in another ongoing activity (e.g., Janczyk, Durst, & Ulrich, 2017; Miller & Durst, 2014). I suggest that this relation be made explicit in the manuscript, since one critical aspect of such paradigms is the temporal spacing of events, which determines the balance between sustained readiness and periodic re-engagement. Finally, constructing a study that genuinely represents real driving is not only a technical issue but also a deeper philosophical problem. Brunswik (1955), in his notion of representative design, already noted that every experimental model requires a decision about which elements of the natural situation to preserve and which to emphasise at the expense of others that are simplified or muted. This choice defines what kind of knowledge an experiment can produce. To put it simply, different designs resemble different types of maps: one may depict political borders in colour, another geological layers through shading, another may adopt a topological rather than a topographical representation. Each offers a partial but purposeful view of the same reality. I would suggest elaborating on this point in the discussion, as such a reflection would make the paper intellectually stronger and conceptually deeper.

Brunswik, E. (1955). Representative design and probabilistic theory in a functional psychology. Psychological Review, 62(3), 193-217. doi:10.1037/h0047470

Janczyk, M., Durst, M., & Ulrich, R. (2017). Action selection by temporally distal goal-states. Psychonomic Bulletin & Review, 24(2), 467-473. doi:10.3758/s13423-016-1096-4

Miller, J., & Durst, M. (2014). "Just do it when you get a chance": the effects of a background task on primary task performance. Attention Perception & Psychophysics, 76(8), 2560-2574. doi:10.3758/s13414-014-0730-3

(3) Theoretical mechanisms

The study comprises three conditions. In the first (pure) condition, participants performed the primary task and were instructed to brake whenever the secondary, prospective-event signal (the standard red stop sign) appeared. This condition already involves a temporal expectancy mechanism: after each occurrence, readiness rises, is actively maintained for a short period, and then decays gradually across subsequent trials. In the two mixed conditions, additional events (decoy stop-road signs) were interspersed. Although these decoy signals did not require overt braking, they nevertheless required stimulus evaluation and thus engaged the same detection process. In this way, they function conceptually as decoy prospective-event signals. R1 already noted that adding such decoy events among the normal targets increases the rate at which participants must pause, evaluate, and process incoming stimuli, and this is in fact the critical variable at work in the present design. With shorter event-to-event distances, the expectancy window is repeatedly reactivated, keeping participants in a state of continuous readiness for potential braking. This pattern means that the observed response slowing in the mixed conditions has nothing to do with feature-conjunction search or feature-integration mechanisms and does not reflect interference from distractor features. It rather reflects the sustained activation of the braking schema due to the higher frequency of prospective events. In other words, the shorter event-to-event distance keeps participants effectively in a be-ready-for-braking mode. I suggest that the authors reconsider the theoretical rationale of the study not in terms of feature similarity or conjunction complexity, but in terms of the temporal structure of expectancy, effort mobilisation, and the maintenance of active readiness (see Schumann et al., 2022, chap. 4.5, for a theoretical analysis). Elaborating on this mechanism in a revision would substantially deepen the theoretical coherence of the manuscript.

Altmann, E. M. (2002). Memory for goals: an activation-based model. Cognitive Science, 26(1), 39-83. doi:10.1207/s15516709cog2601_2

Altmann, E. M. (2002). Functional decay of memory for tasks. Psychol Res, 66(4), 287-297. doi:10.1007/s00426-002-0102-9

Schumann, F. et al. (2022). Restoration of attention by rest in a multitasking world: Theory, methodology, and empirical evidence. Frontiers in Psychology, 13, 867978. doi:10.3389/fpsyg.2022.867978 

Reviewers' comments:

Reviewer's Responses to Questions

**Comments to the Author**

Reviewer #1: All comments have been addressed

2. Is the manuscript technically sound, and do the data support the conclusions?

Reviewer #1: Yes

3. Has the statistical analysis been performed appropriately and rigorously?

Reviewer #1: Yes

4. Have the authors made all data underlying the findings in their manuscript fully available?

Reviewer #1: No

5. Is the manuscript presented in an intelligible fashion and written in standard English?

Reviewer #1: Yes

Reviewer #1: (No Response)

**Do you want your identity to be public for this peer review?** For information about this choice, including consent withdrawal, please see our Privacy Policy

Reviewer #1: No

---

## [Author Response · Author response to Decision Letter 2]

17 Nov 2025

We have addressed all comments from the Editor point by point in the uploaded Word file entitled “ResponseToEditor_PLOS.” All corresponding changes have been highlighted in the revised manuscript.

---

## [Editor Report · Decision Letter 2]

18 Nov 2025

Prepared for the expected but unready for the unexpected: unmet distractor expectations slow braking responsiveness but improve lane-keeping precision in a virtual driving simulation

PONE-D-25-01476R2

Dear Dr. Massironi,

We’re pleased to inform you that your manuscript has been judged scientifically suitable for publication and will be formally accepted for publication once it meets all outstanding technical requirements.

Final editor comments: I have read the final version of your manuscript and your responses. Conceptually, I am still not fully convinced; most of my original concerns still remain present. At the same time, I am impressed by the clarity and the way of arguing with which you defend your work, so this means, I think we have reached the point where continuing the exchange would not add much further. Nothing is perfect, the paper makes its contribution, and time will show how it will be received. I am therefore satisfied to recommend acceptance. My congratulation!

Kind regards,

Michael B. Steinborn, PhD

Section Editor

PLOS ONE
---

## [Editor Report · Acceptance letter]

PONE-D-25-01476R2

PLOS One

Dear Dr. Massironi,

I'm pleased to inform you that your manuscript has been deemed suitable for publication in PLOS One. Congratulations! Your manuscript is now being handed over to our production team.

Kind regards,

on behalf of

Dr. Michael B. Steinborn

Section Editor

PLOS One